# Immobilization of Microbes for Biodegradation of Microcystins: A Mini Review

**DOI:** 10.3390/toxins14080573

**Published:** 2022-08-22

**Authors:** Jiajia Zhang, Jia Wei, Isaac Yaw Massey, Tangjian Peng, Fei Yang

**Affiliations:** 1Hunan Provincial Key Laboratory of Clinical Epidemiology, Xiangya School of Public Health, Central South University, Changsha 410078, China; 2Hunan Province Key Laboratory of Typical Environmental Pollution and Health Hazards, School of Public Health, Hengyang Medical School, University of South China, Hengyang 421001, China; 3The Key Laboratory of Ecological Environment and Critical Human Diseases Prevention of Hunan Province, Department of Education, School of Basic Medical Sciences, Hengyang Medical School, University of South China, Hengyang 421001, China

**Keywords:** microcystins (MCs), microbial immobilization, biodegradation

## Abstract

Harmful cyanobacterial blooms (HCBs) frequently occur in eutrophic freshwater ecosystems worldwide. Microcystins (MCs) are considered to be the most prominent and toxic metabolites during HCBs. MCs may be harmful to human and animal health through drinking water and recreational water. Biodegradation is eco-friendly, cost-effective and one of the most effective methods to remove MCs. Many novel MC-degrading bacteria and their potential for MCs degradation have been documented. However, it is a challenge to apply the free MC-degrading bacterial cells in natural environments due to the long-term operational instability and difficult recycling. Immobilization is the process of restricting the mobility of bacteria using carriers, which has several advantages as biocatalysts compared to free bacterial cells. Biological water treatment systems with microbial immobilization technology can potentially be utilized to treat MC-polluted wastewater. In this review article, various types of supporting materials and methods for microbial immobilization and the application of bacterial immobilization technology for the treatment of MCs-contaminated water are discussed. This article may further broaden the application of microbial immobilization technology to the bioremediation of MC-polluted environments.

## 1. Introduction

Harmful algal blooms (HABs) in fresh water and marine ecosystems have become a severe threat to public health due to their global occurrence, intensity, and duration [1,2]. The HABs may destroy the water ecosystem, reduce water quality, and release cyanotoxins [3]. Microcystins (MCs) are the most widely distributed and abundant toxins associated with freshwater HABs [4,5,6,7]. MCs are cyclic heptapeptide hepatotoxins, of which more than 270 isoform types have been isolated [8]. People are exposed to the MCs in various ways, including direct contact, inhalation, and ingestion [9,10]. There have been some cases of lethal or severe poisoning in humans after acute cyanotoxins exposure; these include the 1988 epidemic of gastroenteritis in Brazil caused by ingestion of water containing cyanotoxins, resulting in 88 deaths [11]; and in 1996, in Caruaru, Brazil, 52 dialysis patients die from dialysis treatment with MC-contaminated water [12]. MCs have been confirmed to have multiple organ toxicity [13,14,15,16,17]. Common signs and symptoms of acute MC poisoning include abdominal pain, fever, headache, dermatitis, and increased liver enzymes [18]. In general, MC-LR can depress the activities of protein phosphatases 1 and 2A and induce reactive oxygen species (ROS) production, followed by the destruction of the cell cytoskeleton, eventually leading to liver cell necrosis and apoptosis [19,20,21]. Also, MC-LR is classified as a possible human carcinogen (Group 2B) [22]. A study revealed that people living on Meiliang Bay of Lake Taihu are exposed to high levels of MCs and are under serious threat of liver injury [23]. To minimize the health risk of MCs, the World Health Organization (WHO) has set a preliminary guideline for microcystin concentrations in drinking water of 1 µg per liter based on the concentration in whole water as ingested and assuming that an adult consumes 2 L per day [24]. The toxicity of MCs and their negative impacts on the environment and socio-economy have drawn extensive attention from the scientific community [25]. It is urgent to search for appropriate strategies to eliminate these toxins. Physical, chemical, and biological methods can be used to remove MCs from water, and biodegradation is a promising candidate because it is eco-friendly and cost-effective [26,27,28,29]. However, biodegradation of MCs using free bacteria cells may encounter many challenges, such as low operational stability and difficulties in recovery and reusing [30]. Immobilization of MC-degrading bacteria was proposed to overcome these problems. This review article describes the biodegradation of MCs and is focused on the microbial immobilization technology and current application of microbial immobilization in MC removal.

## 2. Biodegradation of MCs

The general structure of MCs is cyclo-(D-Ala^(1)^-R1^(2)^-D-isoMeAsp^(3)^-R2^(4)^-Adda^(5)^-D-isoGlu^(6)^-Mdha^(7)^-), where R1 and R2 represent a pair of highly variable L-amino acids [31,32] (Figure 1). As a group of cyclic heptapeptides, MCs have a cyclic structure, which makes them stable under harsh conditions such as high temperature, extreme pH, and sunlight [33]. Considering the health risk of MCs, efficient, convenient, and cheap methods capable of detoxifying MCs are urgently required. Conventional strategies used to manage and control MCs include physical and chemical methods. Although physical and chemical methods have been proven to be effective in removing MCs, they are limited by high cost, elevated energy consumption, low removal efficiency and reactive byproducts [34,35,36]. In recent years, bacterial degradation of MCs has been recognized as an environmentally friendly method, and various bacteria with the ability to degrade MCs have been isolated and studied [37,38,39,40,41]. The biodegradation activity of free MC-degrading bacteria may be reduced when temperature and pH are changed [42,43,44]. Therefore, microbial immobilization technology is proposed as a strategy to improve operational stability.

Applying microbial immobilization technology offers distinct advantages over free bacterial cells [45,46]. For instance, (i) bacteria cells immobilized in supporting materials can keep cells in high biomass, make them easy to separate, and be protected against the poor natural conditions; (ii) immobilized microorganisms can form a self-circulation and stable biological systems to avoid contamination by other microorganisms; and (iii) immobilized cell systems could eliminate the cell washout problems at high dilution rates. Due to these benefits, the application of microbial immobilization technology to remove contaminants from water has attracted widespread attention.

Microorganism immobilization technology has been widely researched and explored in water contaminants’ treatment [47,48,49]. In the past few decades, kinds of bacteria capable of degrading MCs have been isolated from natural conditions [44,50,51,52,53]. The feasibility of using MC-degrading bacteria in immobilized states [30], such as within biofilms in water treatment plant (WTP) filters, may serve as an attractive treatment barrier for MCs.

## 3. Microbial Immobilization and Its Application for MCs Degradation

### 3.1. Microbial Immobilization

Microbial immobilization is a widespread natural phenomenon existing in the environment. Ho et al. found that biodegradation occurred in a sand filter where bacteria are present in the biofilm [54]. Biofilms are surface-attached microbial communities consisting of multiple layers of cells embedded in hydrated matrices, which are common and simple immobilized systems that can remove contaminants from drinking water naturally [55]. Biofilm occurring in natural conditions encourages humans to utilize it for their services. Evidence has shown that biofilm formed by known MC-degrading bacteria could be used in several different reactor systems, for example, a recycled membrane biofilm reactor and a fluidized bed biofilm reactor [56,57]. Biodegraders immobilized as biofilm systems have many advantages such as being easy to operate, mild, and cheap.

Immobilization is the physical or chemical confinement or localization of biocatalysts (enzymes or intact cells) to a specific region of space with the preservation of some desired catalytic activity [58,59]. Microorganism immobilization is achieved by attaching to or embedding in organic or inorganic water-insoluble materials [60]. Enzymes and bacteria with contaminant degradation ability could be immobilized by different carriers and immobilization methods [61]. It has been reported that several pollutants, including MCs [30], heavy metals [62], polychlorinated biphenyls [63], dyes [64] and phenols [65], could be eliminated by immobilized bacterial cells. Microbial cell immobilization aims to increase biomass, reuse the biological components, and protect them from extreme conditions by restricting cell mobility to a limited region. Therefore, microbial immobilization technology is considered as a promising method for wastewater treatment.

### 3.2. The Carriers of Microbial Immobilization

The carrier plays a significant role in the microbial immobilization system. It not only directly affects the viability of microorganisms, but also affects the treatment efficiency of the MC removal process. The carrier with good performance should have certain characteristics [66]. It should not be biodegraded by the immobilized microorganisms and should not have toxicity on the microbial cells. It should provide high biological and chemical stability and a large capacity for microbial cells. Further, it should be inexpensive, easy to handle and regenerate. The purpose of all carriers to be utilized is to provide a place for biological reactions and establish a safe condition, which keeps the free bacteria cell native biological activity [67].

As previous studies have mentioned, carriers and supporting material can be divided into three types: inorganic carriers, organic carriers, and composite carriers [68]. A variety of inorganic carriers or supporting materials have been successfully used in immobilization, including porous ceramic and glass, carbon-based materials, diatomite, and quartz sand. For example, in the study by Zheng et al., *Pseudomonas putida* immobilized on activated carbon (AC) performed efficiently in the degradation and adsorption of anionic dyes [69]. In another study, the porous ceramic was used as a supporting material used for the attachment of fresh activated sludge to treat oilfield polluted water [70]. In general, inorganic substrates have the advantage of being easy to find and having excellent mechanical strength. Organic carriers are divided into nature carriers and synthetic polymers. Synthetic polymers, including alginate, chitosan, and polyvinyl alcohol (PVA), are more widely used in immobilization. Organic materials-based carriers have a larger capacity for bacteria loading and provide higher diffusion rates. Bai et al. [71] used crosslinked PVA as a carrier to immobilize nitrifying bacteria, and Nie et al. [48] immobilized *Rhodococcus qingshengii* strain FF on the surface of polyethylene. Quartz sand is a common carrier often used in biological filtration (BF), which has received extensive attention as an optimized method for sand filtration [72]. However, sand still has the disadvantage of low adsorption capacity and cannot protect bacterial cells from environmental pollution. Compared with sand, alginate has more obvious advantages in immobilized bacteria, including good permeability, transparency, and protection [73]. Alginate can encapsulate microbial cells, thereby separating them from the natural environment, while also allowing gas and liquid molecules to diffuse into the interior [74]. Shaharuddin et al. [75] use sodium alginate to immobilize and microencapsulate *Lactobacillus rhamnosus* NRRL 442 to enhance cell survivability after heat exposure at 90 °C for 30 s.

Composite materials composed of two or more organic or inorganic materials are superior to inorganic or organic carriers because they combine a variety of advantages of materials. As an example, Ren et al. [76] used activated carbon fiber–sodium alginate (ACF-SA) to immobilize the MC-degrading bacterium *Sphingopyxis* sp. YF1 cells, obtaining an immobilized system (ACF-SA@YF1). In addition to the capability of MC-RR degradation, the system also has satisfactory cycle stability. The MC-RR removal efficiency was 70.38% after seven cycles and 78.54% of initial activity after 20 days of storage.

To make immobilization technology sound and economically feasible, it is necessary to optimize and develop efficient carriers. With the development of new nanomaterial technology, nanomaterials show their excellent properties and emerge in a composed carrier to enhance immobilization [77]. The superiorities of nanomaterials are unique characteristics such as their small size and physicochemical properties [78]. Iron oxide nanomaterial-based immobilization technology is widely used in wastewater treatment owing to its excellent chemical inertness and favorable biocompatibility [79]. Three common types of iron oxides employed for immobilization are magnetite (Fe_3_O_4_), maghemite (γ-Fe_2_O_3_), and hematite (α-Fe_2_O_3_) [80,81]. Bestawy et al. [82] used Fe_3_O_4_ nanoparticles (Fe_3_O_4_ NPs) to immobilize *Enterobacter cloacae* and *Pseudomonas otitidis* to treat wastewater contaminated by petroleum hydrocarbon. The highest removal efficiency of total petroleum hydrocarbon was 85%, and the residual concentration was 60 mg/L after 4 h of treatment by Fe_3_O_4_-bacteria composite. The results showed that Fe_3_O_4_-bacteria composite could keep higher biomass than Fe_3_O_4_-free bacteria. In the study by Zamani et al. [83], Fe_3_O_4_ nanoparticles (NPs) were used to enhance the biodegradation of paclitaxel by *Citrobacter amalonaticus* immobilized on alginate gel beads. Results demonstrated that supplementation of sodium alginate gel beads with 5, 10, and 20 mg/L of Fe_3_O_4_ NPs enhanced biodegradation efficiency to 66.0%, 80.0%, and 78.0%, respectively. In another study, Fe_3_O_4_ NPs were employed to confer magnetism to the immobilized *Bacillus methylotrophicus* ZJU, which could facilitate the recollection of the cells from water after their application [84]. Furthermore, Fe_3_O_4_ NP-based composite carriers were also utilized to immobilize enzymes, including lipase [85], laccase [86], and pectinase [87].

A few studies have focused on the application of emerging membrane technologies, such as the membrane biofilm reactor (MBfR) in water treatment. The membrane acts as a carrier to provide a substrate for bacteria to form a biofilm. The surface characteristics of reverse osmosis membranes (ROM) are more suitable for microorganisms to form biofilm due to the contact with organic sludge in seawater filtration. Jesús et al. [88] used discarded RO membranes as support materials to immobilize *Sphingopyxis* sp. strain IM-1, and built a system named Recycled-Membrane Biofilm Reactor (R-MBfR). This system can recycle the discarded RO membranes, and also provides a new way for the treatment of MCs-polluted water.

### 3.3. The Methods of Microbial Immobilization

Microbial cell immobilization is a technique that immobilizes microbial cells on carriers or supporting materials using physical or chemical methods [89].

The physical methods binding microbial cells on carriers are irreversible, including entrapment and encapsulation [66], as shown in Figure 2. The entrapment method is capturing particles or cells within a support matrix or inside a hollow fiber, which provides a barrier around microbial cells to protect cells from external conditions and decrease the leakage of cells [66,90]. Various types of carriers with porous structures are used in entrapment, such as chitosan, alginate, polyester, polystyrene, and polyurethane. Encapsulation is a special type of entrapment in which biological components are enveloped within various forms of semi-permeable membranes. The main advantages of the two types of physical methods are that carriers offer barriers around microbial cells and allow pollutants to diffuse into the matrix.

The chemical methods for the immobilization of cells on carriers are adsorption and covalent binding [66], as shown in Figure 2. Adsorption is considered as the simplest and oldest reversible method in which bacterial cells directly adhere to the surface of water-insoluble carriers through van der Waals interactions, ionic forces and hydrogen bonds [91]. A bacterial biofilm is a typical form of bacterial immobilization through adsorption. The governing advantages of adsorption are its mildness, easy operation, and low cost, and it allows microorganisms to make direct contact with contaminants. Covalent binding is based on the formation of a covalent bond by the existence of chemical additives. Covalent binding is widely used to immobilize enzymes, for example, Zhu et al. [92] used covalent binding to immobilize porcine pancreatic lipase on the carboxyl functionalized silica-coated magnetic nanoparticles. However, the biocatalyst activity and viability could lose due to the toxicity of coupling agents.

The microorganism can naturally immobilize in extracellular polymeric substance (EPS) matrices to form complex structural and functional aggregates on some submerged surfaces, such as periphyton biofilms (PBs), which is an important natural immobilization form of bacterial cells used in wastewater treatment [93,94,95]. Ho et al. provided evidence that the immobilization principle is already found in a sand filter, as bacterial samples collected along sand covered by a biofilm of MCs utilizing bacteria that help degrade MCs in wastewater. This indicated that MC-degrading bacteria from algal blooming water could form biofilms and effectively degrade MC-LR and MC-LA under environmentally relevant conditions in the sand filter [96]. Although indigenous MC-degrading bacteria can form biofilm within a sand filter during wastewater treatment, the low biomass of MC-degrading bacteria, the competition of non-MC-degrading bacteria, and the long-time adaptation period limit the MC degradation efficiency. Therefore, based on PB-driven biodegradation, biological filtration (BF) appears as an attractive application measure for MCs removal and garners particular attention [97]. The biofilm system has been successfully established in the laboratory and applied to the continuous degradation of MCs in wastewater. Another study by Ho et al. has shown that columns packed with sand can establish a biofilm and acclimate to MCs in quite a short time. Moreover, the results showed that pre-exposure to MCs was able to shorten or even eliminate the lag period prior to degradation commencing [54]. Using a known cultured MCs-degrading bacterial strain (*Sphingomonas* sp. (MJ-PV)) co-immobilized with indigenous bacterial from river water could facilitate MC-LR levels to be reduced to below 1 µg/L, according to microcosm experiments [98].

### 3.4. Present Understanding of Microbial Immobilization for MCs Degradation

Bioremediation is considered to be one of the effective ways to treat MCs in water, but it takes a long time for indigenous bacteria to adapt to the environment and MCs decompose slowly [99]. Microbial immobilization is helpful to improve the biodegrading efficiency and develop the application of indigenous bacteria. Table 1 is listing out application of immobilized microorganisms for removal of MCs.

Wu et al. [30] and Ren et al. [76] investigated the whole-cell immobilization of *Sphingopyxis* sp. YF1. Ren et al. studied the degradation of MC-RR by *Sphingopyxis* sp. YF1 immobilized active carbon fiber optimized–sodium alginate (ACF-SA). ACF is more suitable for the immobilization of microorganisms owing to its good capacity, high chemical durability, and mechanical strength. The result of this study showed that the degradation rate of ACF-SA@YF1 could achieve 0.76 µg/mL/h at 30 °C and pH 7.0, respectively. As described by Wu et al., a novel bio-functionalized composite carrier synthesized by chitosan-grafted Fe_3_O_4_ magnetic particles (Fe_3_O_4_@CTS) was used to immobilize *Sphingopyxis* sp. YF1. The degradation efficiency of MC-LR by Fe_3_O_4_@CTS/*Sphingopyxis* sp. YF1 was high and not affected by the change of pH (6–9) and temperature (25–35 °C). Interestingly, the MC-LR degrading efficiency was enhanced after reusing and recycling and reached 1.5 µg/mL/h in the sixth cycle. The utilization of Fe_3_O_4_ magnetic particles makes *Sphingopyxis* sp. YF1 easily recycled. Three MC degradation-related genes (*mlrA*, *mlrB*, and *mlrC*) were obtained in this study. Meanwhile, degradation intermediates of MC-LR by the Fe_3_O_4_@CTS/*Sphingopyxis* sp. YF1 were analyzed by HPLC. The HPLC chromatograms showed that tetrapeptide and Adda were the intermediates of MC-LR degradation by the *Sphingopyxis* sp. YF1 [37]. Using the two types of composite carriers, immobilization of YF1 achieved high degrading activity, environmental friendliness, simple preparation, low consumption, outstanding stability, and recyclability.

Hai et al. [100] implemented a study on the immobilization of *Ralstonia solanacearum* on the carbon nanotubes (CNTs). The *Ralstonia solanacearum* cells could easily adhere to the surface of CNTs. Compared with free cells, *Ralstonia solanacearum* immobilized on CNTs exhibited higher removal rates of MC-LR and MC-RR in 3 and 9 h. CNTs as a supporting material could also absorb amounts of MC-RR and LR and provide a place for biodegradation, which was responsible for the increase in removal efficiencies of MC-RR and MC-LR from water.

Tsuji et al. [101] immobilized a strain B-9 with polyester (Fabios) and polyethylene glycol gel (PEG) by adhering and entrapping methods, respectively. Results revealed that B-9 immobilized using the adhering method showed a higher degradation rate than the entrapped method. The MC-RR degradation efficiency of adhesively immobilized B-9 reached 100% in the first trial. The repeated use caused a gradual decrease in the efficiency and decreased to 65% in the fourth trial. Results of this study indicate that 600 mg MC-RR was almost totally degraded by B-9 immobilized with polyester (Fabios) in a 3 L bioreactor. The MC-RR removal efficiency of immobilized B-9 with polyester (Fabios) could be higher than 80% after almost 2 months of continuous bioreactor operation.

It has been documented that the linearization reaction catalyzed by metalloproteinase called microcystinase (MlrA) is a crucial step in MCs degradation, which can significantly reduce the toxicity of MCs. Dziga et al. [102] investigated the MC degradation efficiency of genetically engineered bacteria (*E. coli* BL21_MlrA) immobilized in alginate. The results showed that the immobilized cells may also express relatively high MlrA activity for a few weeks. A column packed with alginate entrapped *E. coli* BL21_MlrA cells could eliminate MC-LR at a rate of 100.3 µg/L/h, and such degradation ability was stable over at least 2 days.

Somdee et al. used abiotic surfaces made of polystyrene, polyvinyl chloride plastic, glass, stainless steel, and ceramic to determine biofilm formation [107]. They demonstrated that the NV3 isolate attached most effectively to ceramic, followed by PVC, polystyrene, stainless steel, and glass coupons. In their study, an internal airlift loop ceramic honeycomb support bioreactor (IAL-CHS bioreactor) was used for removing MCs. In the batch experiment, NV-3 degraded [Dha^7^] MC-LR at an initial concentration of 25 µg/mL at 30 °C in 30 h, whereas in the continuous-flow experiment, NV-3 degraded the same concentration of [Dha^7^] MC-LR in 36 h with hydraulic retention time (HRT) of 8 h.

Pratik et al. [57] used K1 Kaldness media as the bio-carrier to immobilize known MC-LR degraders: *Arthrobacter ramosus* and *Bacillus* sp., and the heterogeneous bacterial community (HBC) present in the sedimentation-unit sludge as a background matrix. In the study, a fluidized bed biofilm reactor (FBBR) was employed to remove MC-LR from drinking water sludge. The bioreactor was inoculated with *Arthrobacter* (RA) and *Bacillus.* (RB). Despite the sloughing events, it showed good long-term (over 300 days) biofilm development. Bioreactor RA and RB showed higher MC-LR removal (93.75% and 90.24%) than the bioreactor with just the sludge mixture (78.37%: without MC-LR degrading bacteria: RC) at an initial concentration of 50 µg/mL MC-LR in each of the reactors.

Immobilization of known MC-LR degraders with a native bacterial community can be used to enhance the removal efficiency of MCs in a sand filter. In the study by Pratik et al. [104], the agricultural waste residue in the form of deinking sludge (DSF), hemp fiber (HFF), and paper-pulp dry sludge (PPF) were used as carriers to form a biofilm of *Arthrobacter*
*ramosus* (NRRLB-3159) with native bacterial strains. The results revealed that agro-residue as the support material enhanced the pathogenic bacteria entrapment due to their less pore volume behavior as compared to the sand particle. DSF, HFF, and PPF achieved MC-LR removal of 87 ± 14%, 82 ± 7% and 78 ± 4%, respectively.

Kumar et al. [105] immobilized known MC-LR degraders (*Arthrobacter*
*ramosus* and *Bacillus* sp.) with native bacterial species isolated from a drinking water treatment plant (DWTPs) in sand coated with reduced graphene oxide (rGO). The results showed that the highest MC-LR removal of 91% was obtained under the biodegradation phase (stage 3) using rGO-coated sand, which showed an increase of 47.2% in MC-LR removal compared to the physical adsorption phase.

Somdee [106] demonstrated that *Novosphingobium* sp. KKU15 immobilized on the sand in columns effectively degraded [Dha^7^] microcystin–LR. They observed that [Dha^7^] MC-LR with a final concentration of 5 µg/mL in MC-contaminated water was completely removed within 7 days in the slow sand filter inoculated with *Novosphingobium* sp. KKU15.

In another study, a bioreactor packed with plastic medium for *Novosphingobium* sp. KKU25s immobilization was then used to treat fresh synthetic wastewater plus [Dha^7^]MC-LR at a concentration of 25 µg/L [103]. In the batch experiment, immobilized KKU25s completely degraded [Dha^7^]MC-LR at 30 °C within 24 h, whereas in the continuous flow experiment, KKU25s degraded the toxin at the same concentration within 36 h.

Other supporting materials have also been investigated as carriers for the growth of biofilm. The reusing of the discarded RO membranes for biofilm immobilization is capable of selectively removing MC-LR in a short time (from 2 mg/L below to the detection limitation within 24 h) [88].

As mentioned above, biofilm is a significant form of microorganism immobilization and widely used in MC removal. Factors affecting the information of biofilm could also affect the MC degradation efficiency. The most important factor affecting the attachment of MC-degrading bacteria is the characteristics of the supporting material. Immobilization of microcystin-degrading bacteria from algal blooming water on sand broadens the application range of bacteria, but the specific surface area of sand is small and its adsorption capacity to bacteria is low [108]. Granular activated carbon (GAC) has been documented as a better supporting material in filter for biofilm formation than sand, because GAC offers a much rougher structure rich in crevasses and ridges and protects newly-attached microbes from shear forces that impact the attachment [109]. Some studies also have focused on optimizing sand to increase its specific surface area and adsorption capacity. Agro-industrial residues as support materials along with sand have been successfully used to immobilize three known MC-LR degraders with native bacteria, increasing the biomass and prolonging the bioactivity of the sand filter model [104]. The inherent properties of MC-degrading bacteria could also affect their attachment to the surface of carriers. Take an example, Ho et al. [110] studied the biofilm formation of two microcystin-degrading bacteria, LH21 and ACM-3962, on a polystyrene surface. The result showed that LH21 formed a biofilm on the polystyrene surface. Conversely, ACM-3962 was unable to form a biofilm on the polystyrene surface. As described by Bourne et al., MJ-PV complete degradation of microcystin LR may require the presence of a consortium of bacteria in a biologically active filter [98]. This phenomenon may indicate that MC degradation processes are the interaction between different bacterial communities and MC-degrading bacteria that are immobilized in biofilm, although the MCs removal efficiency mainly depends on the seeded MC-degrading bacteria.

## 4. Summary

HCBs in the fresh water system have become a growing global public problem. MCs from HCBs may enter drinking water and threaten the health of humans and animals. MCs are stable under natural conditions, and many studies have reported physical and chemical methods to eliminate MCs. However, these methods are limited by many factors, including complex operation, poor efficiency, high cost, and secondary pollution. Biodegradation of MCs by isolated bacteria has been confirmed to be efficient, economical, and environment-friendly. Compared to free MC-degrading bacteria cells, the immobilization cell system was found to have excellent advantages in the efficiency and continuous treatment of MCs-polluted water bodies. However, the practical, industrial and application of immobilized MC-degrading bacteria in the large-scale in-situ MC-decontamination is limited due to the complex composition of wastewater, complicated operation, and sophisticated construction. Accordingly, to avoid these disadvantages, further studies should consider dealing with the following challenges. First, the supporting material used for MC-degrading bacteria immobilization plays a significant role in the engineering application. However, available supporting materials used for MC-degrading bacteria immobilization are limited. Therefore, it is necessary to develop more efficient, inexpensive, and capacity-supporting materials to achieve high decomposing efficiency. A well-performing stable carrier should have the characteristics of excellent biocapacity, high specific surface area, and a porous structure. Nanoparticle-based materials simplify the immobilization on a large scale without using surfactants and toxic reagents and have been proven to have the capability to improve the efficiency of immobilized cells. Second, immobilization methods are also considered as an important factor affecting the removal efficiency. The application of immobilization cells in conditions requires excellent activity. Hence, methods should focus on the enhancement of the biodegrading-activity of bacteria cells. Adsorption may be an attractive method because it is mild, quick, simple, inexpensive, effective, and it does not require chemical additives. Third, selecting proper MC-degrading bacteria or enzymes could also make the immobilization system achieve higher efficacy. Genetically engineered MC-degrading bacteria may become an alternative candidate with significantly enhanced activity toward MCs. Enzymes have higher biodegrading-activity than bacteria. However, there are few reports on the removal of MC by immobilized MC-degrading enzymes, so it is of great significance to study immobilized MC-degrading enzymes in further. Last, the appropriate cell/carrier ratio also plays an important role in immobilization, because high cell/carriers may lead to overcrowding and a low cell/carrier ratio may result in decreased degradation efficiency. In addition, a proper cell/carrier ratio implies a better control of costs. A new topic of further research is molecular mechanisms on the degradation of MCs by immobilized systems. According to available literature, information about molecular mechanisms on the degradation of MCs by immobilized MCs-biodegradation bacteria is relatively limited, and should be explored to understand the real application of immobilization by further research. In response to these points, immobilized cell technology is a promising topic for further research.

## Figures and Tables

**Figure 1 toxins-14-00573-f001:**
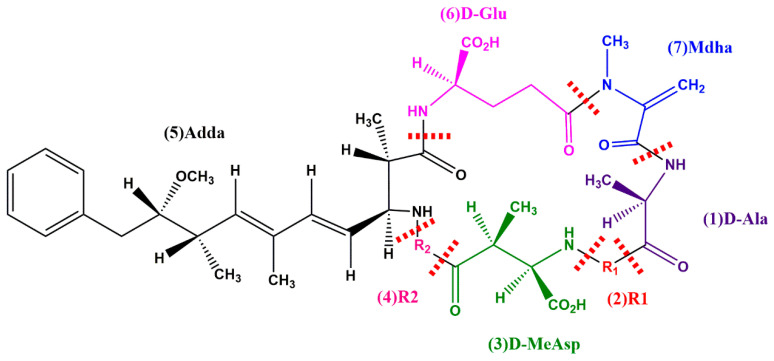
General chemical structure of MCs. 1–7 signify seven amino acid residues. R1 and R2 in positions two and four are highly variable L-amino acids.

**Figure 2 toxins-14-00573-f002:**
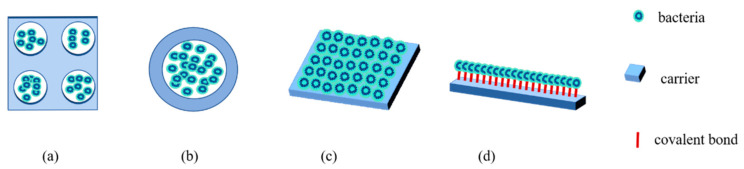
Types of immobilization. (**a**) Entrapment; (**b**) Encapsulation; (**c**) Adsorption and (**d**) Covalent binding.

**Table 1 toxins-14-00573-t001:** Application of immobilized microorganisms for removal of MCs.

Microorganism	Carrier	Microcystins	Initial Concentration (μg/mL)	Degradation Period	Degradation %	Degradation Rate (µg/L/h)	Container	Condition	Reference
*Sphingopyxis* sp. YF1	ACF-SA	MC-RR	12	8 h	100	7.6 × 10^2^	-	30 °C, pH 7.0	[76]
*Sphingopyxis* sp. YF1	Fe_3_O_4_@CTS	MC-LR	10	12 h	100	6.5 × 10^5^	-	30 °C, pH 7.2 (Deionized water)	[30]
*Ralstonia solanacearum*	carbon nanotubes	MC-RR	52.5	within 24 h	100	2.18 × 10^3^	flasks	30 °C, pH 7.0 (PBS)	[100]
		MC-LR	29.5						
B-9	polyester (Fabios)	MC-RR	0.2	24 h	100	1.25 × 10^4^	Aeration bioreactor	25 °C, pH 7.4 (PBS)	[101]
*E. coli* BL21_MlrA	alginate	MC-LR	0.035	-	100	100.3	Column	(freshwater)	[102]
*Sphingomonas* isolate NV-3	ceramic	[Dha^7^]MC-LR	25	30 h	100	8.3 × 10^2^	IAL-CHS bioreactor	30 °C, pH 7.2 (synthetic wastewater)	[103]
*Arthrobacter ramosus* + HBC	K1 Kaldness media	MC-LR	50	6 days	93.75	3.5 × 10^2^	Fluidized bed biofilm reactor	15–19 °C, pH 7.2 (PBS)	[57]
*Bacillus* sp. + HBC	K1 Kaldness media	MC-LR	50	6 days	90.24	34.72			
*Arthrobacter ramosus* (NRRL B-3159) + native bacterial	deinking sludge + sand	MC-LR	0.05	7 cycle study (49 days)	87 ± 14	-	sand filter	pH 6.4 (Lake water)	[104]
	hemp fiber + sand				82 ± 7			pH 6.5 (Lake water)	
	paper-pulp dry sludge + sand				78 ± 4			pH 6.6 (Lake water)	
*Arthrobacter ramosus* + native bacterial species	rGO-coated sand	MC-LR	50	stage (3): 6 cycles	91.4 ± 5.6	-	sand filter	(Lake water)	[105]
*Novosphingobium* sp. KKU15	sand	[Dha^7^]MC-LR	5	7 d EBCT: not mentioned	100	-	slow sand filter	30 °C, pH 7.2 (Mineral salt medium)	[106]
*Novosphingobium* sp. KKU25s	plastic medium	[Dha^7^]MC-LR	0.025	within 24 h	100	1.04	sterile bioreactor	30 °C, pH 7.2 (fresh synthetic wastewater)	[103]
*Sphingopyxis* sp. strain IM-1	RO membranes	MCs	2	24 h	100	83.33	flasks	27 °C, pH 7.2 (Mineral salt medium)	[88]

## Data Availability

Not applicable.

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
