# Peer review of "Immobilization of Microbes for Biodegradation of Microcystins: A Mini Review"

_toxins, 2022, doi:10.3390/toxins14080573_

Round 1
Reviewer 1 Report
This manuscript aims to summarize various types of supporting materials and methods for microbial immobilization and the application of bacterial immobilization technology for the treatment of MCs-contaminated water. However, the discussion lacks some critical thinking. In addition, the manuscript requires extensive editing for grammatical and typographical errors plus the omission of key references within many sections, some sections have absolutely no references at all. Some detailed comments and suggestions are listed as follows:
1. Abstract
L10-12
“Immobilization is the process of restricting the mobility of bacteria using carriers, which have a lot of advantages as biocatalysts compared with free bacterial cells.”
Please change “have” to “has”.
2. Introduction
L23-24
“The blooming of HABs may destroy water ecosystem, reduce water quality and release cyanotoxins [3].”
Please delete “The blooming of”.
3. L25-27
“MCs are cyclic heptapeptide hepatotoxins which have been isolated more than 270 isomer types from HABs [8].”
Please check the grammar.
4. L31-33
“To minimize the health risk of MCs, the World Health Organization proposed 1μg/L as the highest acceptable concentration of MC-LR in drinking water [18].”
Please insert a blank character between “1” and “μg/L”. A blank character should be inserted between the number and the unit. Please check through the entire manuscript.
5. L33-35
“The toxicity of MCs and their negative impacts on the environment and socioeconomic have drawn extensive attention from the scientific community. It is urgent to search for appropriate strategies to eliminate these toxins.”
Please read and cite the following paper.
Challenges of using blooms of Microcystis spp. in animal feeds: A comprehensive review of nutritional, toxicological and microbial health evaluation. https://doi.org/10.1016/j.scitotenv.2020.142319
6. L36-40
“Physical, chemical and biological methods can be used to remove MCs from water, and biodegradation is a promising candidate-because it is eco-friendly and cost-effective. However, biodegradation of MCs using free bacteria may encounter many challenges, such as low operational stability, and the difficulties to recovery and reuse. Immobilization of MC-degrading bacteria was proposed to overcome these problems.”
The references are missing.
7. L57-69
The references are missing.
8. L95-103
The references are missing.
9. L108-110
“For example, in the study of Zheng et al., Pseudomonas putida immobilized on activated carbon (AC) performed efficiently in the degradation and adsorption of anionic dyes [42].”
“Pseudomonas putida” should be italic. Please check all the Latin names of species in the manuscript.
10. L112-113
“In general, common superiorities of inorganic carriers are easy to find and exhibit excellent mechanical strength.”
Please check the grammar.
11. Table 1, Application of immobilized microorganisms for removal of MCs
“Degradation rate”
Some data is missing. Please insert the data from the references or calculate the data according to Initial concentration (μg/mL), Degradation period, Degradation %. Please unify the units of Degradation rate as “μg/L/h”. Please unify the units of Initial concentration as “μg/L”.
Reviewer 2 Report
In this mini review, authors describe the immobilization of microbes for biodegradation of microcystins. Various types of supporting materials and methods for microbial immobilization and the application of bacterial immobilization technology for the treatment of microcystin contaminated water are discussed. The manuscript is clear and well organized. It seems to be suitable for the Journal's aim and purpose. I would like to recommend its publication after some revisions.
Page 3 line 108, please put “Pseudomonas putida” in italics
Page 5 line 211. Replace “1ug/L” with “1μg/L”
Page 6. Table 1. please put Sphingopyxis in italics; “100.3 µg h-1 per” please fix the format and complete the sentence.
Page 7 line 281. Replace “50 µg/ml” with “50 µg/mL”. Same in lines 298 and 303.
Page 8 line 301. Please fix the sentence: “In another study of Phujomjai et al., The bioreactor, complete with the plastic”.
References.
There are several incomplete references such as: line 396 reference number 16 (page numbers?); line 401 reference number 19 (authors?); line 412 reference number 24 (authors?); line 442 reference number 39 (authors?); line 506 reference number 72 (authors?); line 522 reference number 80 (authors?).
The articles 81 (Phujomjai, Y., A. Somdee, and T. Somdee) and 82 are the same. Please fix the numbers in the reference list and the numbering in the manuscript.
Reviewer 3 Report
Immobilization of microbes for biodegradation of microcystins: a mini review
It is advised to mention that The World Health Organization (WHO) has set a preliminary guideline for microcystin concentrations in drinking water of 1 µg per liter based on the concentration in whole water as ingested and assuming that an adult consumes 2 liters per day (https://www.who.int/water_sanitation_health/water-quality/guidelines/chemicals/microcystin/en/).
Also, it is worth mentioning that microcystin-LR are classified as a possible human carcinogen (Group 2B) - International Agency for Research on Cancer, Monographs on the Evaluation of Carcinogenic Risks to Humans.
Write in more details about the risks associated with MCLR.For example, The risk upon exposure includes life-threatening conditions requiring hospitalization, e.g., due to atypical pneumonia followed by dyspnea and liver damage, or an acute hepatic failure requiring a liver transplant. Gastrointestinal symptoms, including abdominal pain, malaise, nausea, vomiting, and diarrhea, were also manifested during these severe poisonings.
Author should also provide a general structure of MC in the first section.
Table 1: Sphingopyxis- change to italics.
At several instances typos were observed with no space between two words, it is advised to go through the manuscript carefully before submitting the revised version.
Table1 shows various immobilization systems used to investigate the efficiency of immobilized cells to remove the MCs from water- Correct this sentence as Table 1 is listing out application of immobilized microorganisms for removal of MCs.
Check the sentence- The result of this study showed that the degradation rate of 229 ACF-SA@YF1 could achieve a rate of 0.76 μg/(mL/h) at 30℃ and pH 7.0, respectively.
Is there any report available on regarding the degradation product of MCs to understand the real application of immobilisation or in general biodegradation at field level? Adding these data or just a paragraph would increase the impact of the review.
Round 2
Reviewer 1 Report
accept